

# "Integrative learning" promotes learning but not memory in older rats

Bin Yin*, Xiao-Rui Wu* and Rong Lian

School of Psychology, Fujian Normal University, Fuzhou, Fujian, China
* These authors contributed equally to this work.

## ABSTRACT

**Background:** We had previously advanced the concept of "Integrative Learning", that is, "under the role of 'meta-learning self', learners actively integrate learning materials to achieve rapid and in-depth understanding of knowledge", and designed an animal behavioral model to compare the effects of "Integrative Learning" (IL) *vs.* "Progressive Learning" (PL) in young rats. It was found that IL is more advantageous than PL. Here, we aim to examine whether the same phenomenon persist in older rats.

**Methods:** Fifteen 12-month-old male Sprague-Dawley (SD) rats were selected as subjects and randomly divided into the IL group and the PL group, and a 14-unit integrative T-maze was constructed for the study. Training and testing procedures contained three stages: the learning stage, the memory retention test stage and the Gestalt transfer learning stage. Data on young rats (1-month-old) from the previous study were also drawn here for comparisons on learning performance.

**Results:** (1) The 12-session learning stage can be divided into three sub-stages as each sub-stage represented the new opening of one third of the whole path in the PL group. There were significant interactions in total errors made between groups and sessions: the PL group had significantly fewer errors during Sub-stage One due to a much shorter path to be learned, however, the IL group's errors made sharply dropped as learning progressed into Sub-stage Two and Three, and were maintained at a significantly lower level than the PL group during Sub-stage Three. (2) When compared with young rats, age had a main effect on the number of errors made—the 1-month-old groups learned overall better and faster than the older groups, whereas the pattern of group differences between the IL and PL learning modes remained consistent across young and older groups. (3) Unlike young rats, during the memory retention test stage and the Gestalt transfer learning stage, the IL group did not perform better than the PL group in older rats.

**Conclusions:** (1) "Integrative Learning" promotes learning but not memory in older rats. (2) Higher-order cognitive abilities that support meta-cognition, long-term retention and knowledge transfer might be deteriorating in older rats.

## INTRODUCTION

Traditional learning focuses on a gradual, from the part to the whole, style. However, this type of learning may not fit well with the new era of information age, in which there may be

Corresponding author
Rong Lian, lianrong1122@126.com

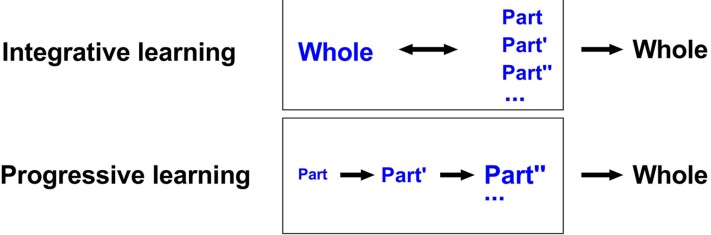

**Figure 1 Conceptual distinction between Integrative Learning and Progressive Learning.** During integrative learning, individuals learn by first encountering and grasping the overall landscape of an area of knowledge, then diving into the details of each part of the area while consolidating the basics and connecting the details to the appropriate position of the overall landscape, and finally forming an integrative map of the area of knowledge, thus exhibiting a "whole-part-whole" pattern of learning. During progressive learning, individuals learn by directly encountering and grasping the basics and details of an area of knowledge, then advancing to a higher level of the area and so forth, and finally forming a finished map of the area of knowledge, thus exhibiting a "part-part-whole" pattern of learning.

too much to be learned and the access to any part of them become easy and available. In order to explore a novel way of learning that may fit better with the needs of the new era, we proposed the concept of "Integrative Learning", which refers to "the process of actively integrating learning materials to achieve an efficient and in-depth understanding and mastery of knowledge under the effect of meta-cognition—it is the psychological process of learning that highly integrates the meta-cognitive and cognitive processes" (*Yin, Wu & Lian, 2020*). In other words, under the guidance of this learning concept, individuals learn by first encountering and grasping the overall landscape of an area of knowledge, then diving into the details of each part of the area while consolidating the basics and connecting the details to the appropriate position of the overall landscape, and finally forming an integrative map of the area of knowledge, thus exhibiting a "whole-part-whole" pattern of learning (Fig. 1). In contrast, non-integrative learning such as progressive learning learn by directly encountering and grasping the basics and details of an area of knowledge, then advancing to a higher level of the area and so forth, and finally forming a finished map of the area of knowledge, thus exhibiting a "part-part-whole" pattern of learning (Fig. 1).

The concept of "integrative learning" was proved to be able to provide more efficient knowledge comprehension and long-term migration in an animal behavioral model using a modified 14-unit composite T-maze (*Yin, Wu & Lian, 2020*), originally designed by Edward Chase Tolman, the famous psychologist who proposed the concept of "cognitive map" (*Tolman, 1948*). Briefly, forty 1-month-old (short as 1 Mo hereafter) Sprague-Dawley (SD) rats, half male and half female, were randomly divided into the "Integrative Learning" (IL) group and the "Progressive Learning" (PL) group. For the IL group, the whole path was clear from the beginning, providing the subjects with opportunities to explore the whole maze without blockage; whereas for the PL group, the whole path was divided into three segments (and the 12-session learning stage was divided into three sub-stages corresponding with the opening of each new segment), providing the subjects with

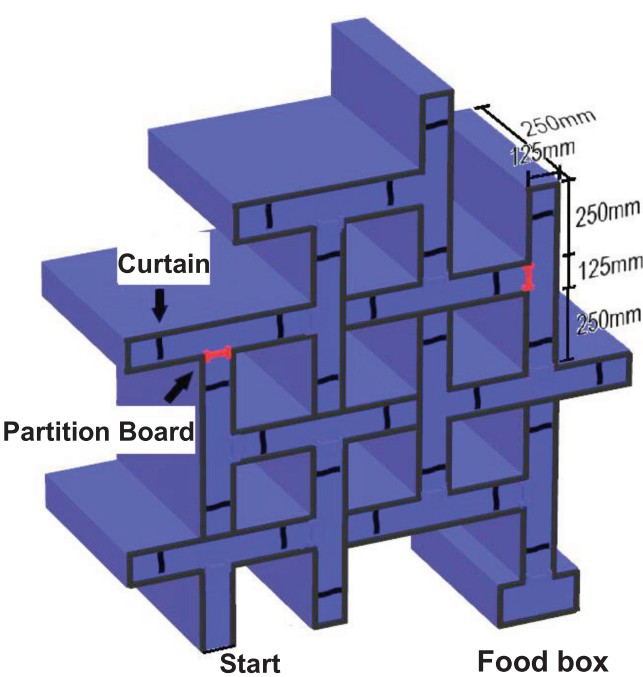

**Figure 2 Schematic drawing of the experimental apparaus.** Basic experimental apparatus, modified from Fig. 1 of *Tolman (1948)*, which material as a whole is now in the public domain. It includes the curtains, the partition boards and the food box (on the left end of the food box there was a rectangular black porcelain bowl). The maze was used throughout the whole study. Of special note is that during the learning stage the partition boards were used to separate the three segments of the maze only for the PL group, and the position of the foodbox was always at the end of the opened segments.

opportunities to explore each segment step by step (Fig. 2). The study found that the IL group initially made a lot of errors due to the difficulty of the task but the errors made soon dropped sharply as learning went on, and maintained at a meaningfully lower level compared with the PL group. Although the study did not find significant differences among groups during the memory retention test 1 week after the learning stage, the IL group performed markedly better during the following test stages designed to evaluate long-term knowledge transfer and adaptation (*Nokes, 2009*). In other words, when the correct route was reversed or modified, the IL group could still find the target with fewer errors and less confusion, suggesting that "Integrative Learning" can better assist knowledge transfer and goal-directed behavior (*Frese & Sabini, 2021*). The study also found that the males benefited more from "Integrative Learning" than the females, possibly due to sex differences in risk taking and/or associative learning in rats (*Jolles, Boogert & van den Bos, 2015*).

Although "Integrative Learning" was found to be more effective in young rats, it is unclear whether the phenomenon still holds in older rats. Aging brought changes to cognitive abilities such as verbal learning, memory, and problem solving (*Arenberg, 1973*; *Kubanis & Zornetzer, 1981*), which might be a result of changes in brain plasticity (*Mahncke, Bronstone & Merzenich, 2006*; *Bloss et al., 2011*), although some argues that aging brains may have remarkable reorganization and plastic adaptation in its own right

(*Holman & de Villers-Sidani, 2014*). In animal models, research has shown environmentally-induced changes in the brains of older rats which were different from those in young rats (*Cummins et al., 1973*). Therefore, it is reasonable to speculate that effects of "Integrative Learning" might be different between young and older rats.

Here, we adapted our original design of "Integrative Learning" (*Yin, Wu & Lian, 2020*) and tested its effects on older rats. A 14-unit composite T-maze was used as the experimental setup and a series of consecutive experiments and analysis were conducted for the purpose of comparing the effects of different learning modes and exploring the intrinsic mechanisms of the learning effect differences on the behavioral level.

## MATERIALS AND METHODS

### Subjects

Fifteen 12-month-old (short as 12 Mo hereafter) male SD rats were randomly divided into two groups: Integrative Learning-12 Mo (IL-12 Mo, $n = 8$) and Progressive Learning-12 Mo (PL-12 Mo, $n = 7$). Their group identity was not known until right before the experiments – when a tail marker was used to mark their group identity individual by individual picked at random—then they kept their group identity till the end of the experiments. All subjects were housed into five cages (three animals per cage) since they were 2 months old purchased from the Experimental Animal Center of Hangzhou Medical College (Hangzhou, Zhejiang, China) with health records. They were housed in a 24-h air-purified clean-grade animal facility with a constant temperature of 22.5 °C and automatic lighting control (lights off at 8:00 and lights on at 20:00). The cages were located on the shelter rack and were randomized once a week to avoid the location position effect. Food was maintained sufficient (50 g per cage per day) and water was available *ab libitum*. All subjects were handled and interacted with for 15 min three times per week until the formal experiment started when they turned twelve months old.

During the formal experiments, rats were fed 36 g per cage in turn when all three rats in one cage finished their trials in order to keep them in a semi-hunger state but each individual's body weight was strictly monitored and maintained above 85% of its free-feeding weight. If significant and sustained loss of body weights were found over a continuous period of time, the subject would be discontinued with experiments and separately housed with dedicated veterinary care. No such subjects were found and thus all subjects were included in the experiments and analysis. During the learning stage, there was no significant differences in body weights between the IL-12 Mo (334.75 ± 9.242 g) and the PL-12 Mo (324.476 ± 9.880 g) groups, $F_{(1,13)} = 0.577$, $p = 0.461$, $\eta^2 = 0.042$. At the end of the experiments, humane euthanasia was performed using the $CO_2$ methods (*Boivin et al., 2017*).

In addition, data from twenty 1-month-old male SD rats (Integrative Learning-1 Mo, IL-1 Mo, $n = 10$; Progressive Learning-1 Mo, PL-1 Mo, $n = 10$) in our previous study were pooled for comparing their learning behavior between the young and the older under different learning modes.

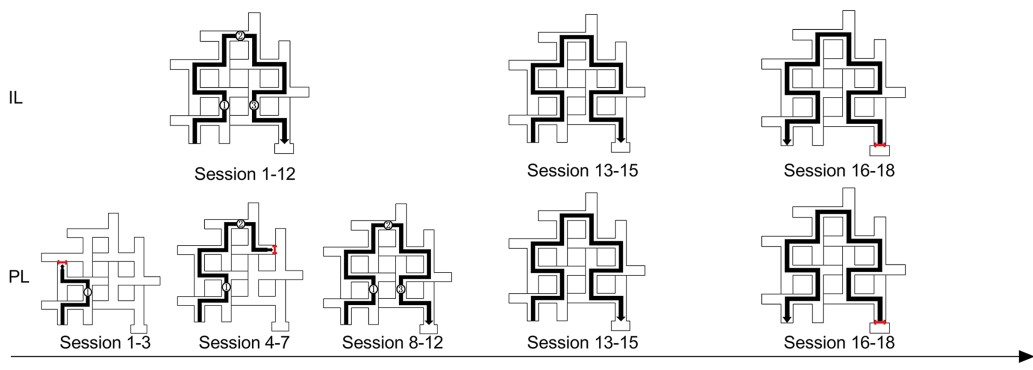

**Figure 3 Experimental task design.** The correct route in the learning stage consists of three segments with identical shapes. The PL group's learning plan was based on the logic of progressive learning: Sessions 1–3 (Sub-stage One) learned the first segment, Sessions 4–7 (Sub-stage Two) learned the first and the second segments; Sessions 8–12 (Sub-stage Three) learned the whole route. The IL group's learning plan was based on the logic of integrated learning: the whole route was kept open throughout the learning stage. During the memory retention test stage and the Gestalt transfer learning stage, the whole route was kept open for both groups. During the Gestalt (reverse) transfer learning stage, the entrance of the original food box was sealed by a partition board, and the subject was placed right in front of the partition board on the original end of the whole route, and the food box was then relocated to the nearest diverting point to the original starting area.

All experiments were approved by the Institutional Animal Care and Use Committee of Fujian Normal University (Approval No. IACUC-20180019) and strictly abided by the ARRIVE 2.0 guidelines (*Du Sert et al., 2020*).

## Apparatus

We used the same apparatus and materials as in our previous study (*Yin, Wu & Lian, 2020*), including a 14-unit composite T-maze (Fig. 2), essential laboratory animal daily care materials, cleaning and protective equipment, video monitoring equipment, video processing and data analysis software, *etc.*, for details see *Appendix S1.2* of *Yin, Wu & Lian (2020)*.

## Study design

The study adopted a two (Learning mode: Integrative Learning/Progressive Learning, or IL/PL) by two (Age: 1/12 Mo) inter-group design, with the number of sessions and the number of total errors made (sum of detection errors and entry errors) as quantitative variables, animal characteristic behavior and tracking routes as qualitative variables. In order to explore the effects of different learning modes, three stages of tasks were designed: the learning stage, the 1-week-later memory retention test stage and the Gestalt (reverse) transfer learning stage (Fig. 3). Detailed task design instructions can be found in the *Appendix S1.3* of *Yin, Wu & Lian (2020)*.

## Experimental procedures

(1) During the adaptation period, the rats were subjected to a handling and interaction procedure for 15 min per cage, three times per week. Food was maintained sufficient (50 g

per cage per day) and water was provided *ab libitum*. Each subject's body weight was recorded during the handling and interaction procedure, while average water and food intake were recorded at the cage level.

(2) The formal experiment consisted of three tasks (Fig. 3). The first was a 12-day learning task (one session per day). On the first day, every subject was limited to 15 min of exploration, and for the 2$^{nd}$ to 12$^{th}$ day, the subjects were immediately taken out after they finished food rewards. The correct route of the maze can be divided into three segments with shape similarity, and thus by using partition boards the PL group was only allowed to explore the first segment during Session 1–3 (Sub-stage One), the first and the second segments during Session 4–7 (Sub-stage Two), and the first, the second and the third segments during Session 8–12 (Sub-stage Three), respectively. The IL group was allowed to explore all three segments from the beginning to the end of the learning stage (Session 1–12). The food reward was placed at the end of each learning segment for the PL group and always at the food-box. One week later, a 3-day memory retention test was performed, followed by a 3-day Gestalt transfer learning task. The condition for memory retention test was identical to Sub-stage Three for both groups; during the Gestalt transfer learning, the starting point and the finale point was reversed so that the subject was placed beside the food-box to start and the food reward was placed at the original starting point. Subjects entered each session in a counterbalanced order and thus the maze was modified accordingly by inserting and removing the partition boards.

(3) The observation record forms were filled throughout the experiments. The *SuperMaze4.0* animal behavior video analysis system (*Shanghai Xinruan*, Shanghai, China) was used to analyze the animal motion track and export the data (including the trajectory map and the heat map). *Microsoft Excel*, *IBM SPSS 18.0*, *GraphPad Prism 8* and *3D drawing* software were used for data analysis and chart plotting. Details of the series of experiments can be found in *Appendices S1.1* and *S1.4* of *Yin, Wu & Lian (2020)*.

The above protocol was not preregistered but was in full accordance to our previous study (*Yin, Wu & Lian, 2020*).

## Statistical methods

For the statistical analysis, learning modes (IL/PL) and age (1/12 Mo) were used as inter-group variables, the number of sessions and the order of sub-stages were used as intra-group variables, and the number of total errors made, the number of days to learning success, whether a certain behavior occurs and the proportion of a certain behavior were used as the dependent variables. The analysis of variance under different combinations of variables was conducted using *IBM SPSS 18.0*. All data were examined for normality and subjected to a Hartley's test to determine whether the data violated assumptions of homogeneity of variance. No significant deviations were found.

## RESULTS

The graphic results of the learning stage, memory retention test stage and Gestalt transfer learning stage are shown in Figs. 4A–4D, respectively. As shown in Fig. 4A and Table 1, during the learning stage, the results of repeated measurement ANOVA with age and

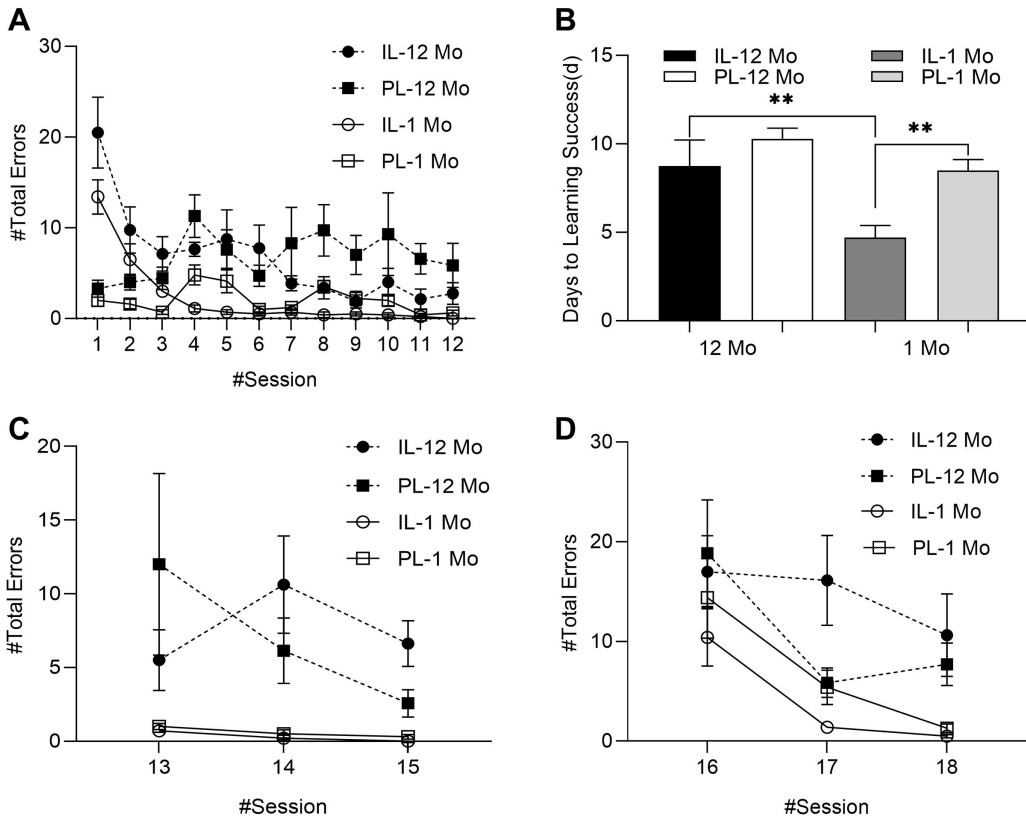

**Figure 4 Comparative results during learning, memory retention test and the Gestalt transfer learning stages.** (A) Learning curve: summary of the number of errors during the learning stage. Among them, "#Total Errors" = "#Entry Errors" + "#Detection Errors", calculated from the beginning of the exploration to the end of the maze. Entry errors refers to the number of times the rat's head and center entered the wrong area (usually through the curtain), while detection errors refers to the number of times the rat's head entered the wrong area (however it did not go through the curtain). (B) The number of days to learning success. Learning success is defined as the number of days from the first learning session to when it can reliably achieve zero errors when exploring the "open segmented route". Among them, the "open segmented route" of the IL group is the whole correct route of the maze, so the number of the days before it can reliably achieve zero errors is recorded as the number of days to learning success; however, the "open segmented route" of the PL group is gradually opened in three stages, so the sum of the number of days before it can reliably achieve zero errors in each stage is recorded as the number of days to learning success. (C) A chart showing the number of total errors during the retest stage 1 week after learning. (D) A chart showing the number of total errors during the Gestalt transfer learning stage. The position of the circle, triangle or column in each chart shows the average value of each group in each task, and the error bar shows the standard error. **$p < 0.01$.

learning mode as intergroup variables and session as an intragroup variable show that the main effect of age was significant in the number of total errors made, and the number of total errors made of the 12 Mo groups is significant more than the 1 Mo groups. At the same time, the interaction between session and learning mode was significant (further simple-effect analysis can be found in Table S1). As shown in Fig. 4B and Table 2, the results of complete random analysis of variance with age and learning mode as intergroup variables showed that the main effects of learning mode and age were both significant in the number of days to learning success. The number of days to learning success of the IL-1

**Table 1 Results of a three-factor repeated measures analysis of variance of the number of total errors made during three learning stages for both 1-month-old and 12-month-old groups.**

|  | SS | DF | MS | F (DFn, DFd) | p | $\eta^2$ |
|---|---|---|---|---|---|---|
| Session | 1,622.522 | 11 | 147.502 | $F$ (11, 341) = 8.307 | 0.000*** | 0.211 |
| Learning mode | 0.114 | 1 | 0.114 | $F$ (1, 31) = 0.004 | 0.953 | 0.000 |
| Age | 2,155.216 | 1 | 2,155.216 | $F$ (1, 31) = 66.827 | 0.000*** | 0.683 |
| Learning mode × Age | 5.992 | 1 | 5.992 | $F$ (1, 31) = 0.186 | 0.669 | 0.006 |
| Session × Learning mode | 2,704.237 | 11 | 245.84 | $F$ (11, 341) = 13.845 | 0.000*** | 0.309 |
| Session × Age | 115.092 | 11 | 10.463 | $F$ (11, 341) = 0.589 | 0.838 | 0.019 |
| Session × Learning mode × Age | 300.878 | 11 | 27.353 | $F$ (11, 341) = 1.540 | 0.115 | 0.047 |

**Note:**
*** $p < 0.001$.

**Table 2 Results of a two-factor complete random analysis of variance on the number of days to learning success.**

|  | SS | DF | MS | F (DFn, DFd) | p | $\eta^2$ |
|---|---|---|---|---|---|---|
| Learning mode | 60.852 | 1 | 60.852 | $F$ (1, 31) = 8.834 | 0.006** | 0.222 |
| Age | 72.791 | 1 | 72.791 | $F$ (1, 31) = 10.568 | 0.003** | 0.254 |
| Learning mode × Age | 10.958 | 1 | 10.958 | $F$ (1, 31) = 1.591 | 0.217 | 0.049 |

**Note:**
** $p < 0.01$.

Mo group was significantly shorter than that of the PL-1 Mo group, while the number of days to learning success of the 12 Mo groups was generally more than that of the 1 Mo groups (further simple-effect analysis can be found in Table S2). However, unlike the 1 Mo groups which showed significant differences between the IL mode and the PL mode all across three learning sub-stages (Table 3), there was no significant differences between the IL mode and the PL mode in the 12 Mo groups except for Sub-stage Three, during which the IL mode showed an advantage by making significantly fewer total errors than the PL mode (Table 4) and exerting much less effort exploring the maze (Fig. S2). Details for interaction and subsequent simple-effect analysis can be found in Tables S3 and S4.

In order to elucidate why in the 12 Mo groups, the differences between the two learning modes were not as vast as in the 1 Mo groups, we further conducted a heatmap analysis for the Sub-stage Two (Session 4–7). This is because during this sub-stage the subjects had been accustomed to the apparatus but had not fully mastered the correct route, providing us the opportunity to see what exactly were different between different age groups. As can be shown in Fig. 5, the IL-1 Mo group spent much less time at the learned routes and mostly at the critical turning points of the unlearned routes, whereas the PL-1 Mo group almost evenly spent their time across the opened segments. In contrast, both the IL-12 Mo

**Table 3 Results of a two-factor repeated measures analysis of variance of the number of total errors made in three learning sub-stages in the 1-month-old groups.**

| | | SS | DF | MS | F (DFn, DFd) | p | $\eta^2$ |
|---|---|---|---|---|---|---|---|
| Session 1–3 | Session | 349.233 | 2 | 174.617 | $F(2, 36) = 14.209$ | 0.000*** | 0.441 |
| | Learning mode | 576.6 | 1 | 576.6 | $F(1, 18) = 41.682$ | 0.000*** | 0.698 |
| | Session × Learning mode | 219.7 | 2 | 109.85 | $F(2, 36) = 8.939$ | 0.001** | 0.332 |
| Session 4–7 | Session | 70.038 | 3 | 23.346 | $F(3, 54) = 5.667$ | 0.002** | 0.239 |
| | Learning mode | 82.013 | 1 | 82.013 | $F(1, 18) = 17.738$ | 0.001** | 0.496 |
| | Session × Learning mode | 46.738 | 3 | 15.579 | $F(3, 54) = 3.781$ | 0.016* | 0.174 |
| Session 8–12 | Session | 40.86 | 4 | 10.215 | $F(4, 72) = 5.927$ | 0.000*** | 0.248 |
| | Learning mode | 51.84 | 1 | 51.84 | $F(1, 18) = 23.492$ | 0.000*** | 0.566 |
| | Session × Learning mode | 25.46 | 4 | 6.365 | $F(4, 72) = 3.693$ | 0.009** | 0.17 |

Note:
* $p < 0.05$.
** $p < 0.01$.
*** $p < 0.001$.

**Table 4 Results of a two-factor repeated measures analysis of variance of the number of total errors made in three learning sub-stages in the 12-month-old groups.**

| | | SS | DF | MS | F (DFn, DFd) | p | $\eta^2$ |
|---|---|---|---|---|---|---|---|
| Session 1–3 | Session | 317.539 | 2 | 158.769 | $F(2, 26) = 4.721$ | 0.018* | 0.266 |
| | Learning mode | 819.432 | 1 | 15.64 | $F(1, 13) = 0.002$ | 0.546 | 0.002 |
| | Session × Learning mode | 437.45 | 2 | 218.725 | $F(2, 26) = 6.504$ | 0.005** | 0.333 |
| Session 4–7 | Session | 117.698 | 3 | 39.233 | $F(3, 39) = 0.970$ | 0.417 | 0.069 |
| | Learning mode | 13.886 | 1 | 13.886 | $F(1, 13) = 0.305$ | 0.590 | 0.023 |
| | Session × Learning mode | 148.364 | 3 | 49.455 | $F(3, 39) = 1.223$ | 0.314 | 0.086 |
| Session 8–12 | Session | 89.378 | 4 | 22.344 | $F(4, 52) = 1.034$ | 0.398 | 0.074 |
| | Learning mode | 441.029 | 1 | 441.029 | $F(1, 13) = 5.485$ | 0.036* | 0.297 |
| | Session × Learning mode | 21.218 | 4 | 5.304 | $F(4, 52) = .246$ | 0.911 | 0.019 |

Note:
* $p < 0.05$.
** $p < 0.01$.

group and the PL-12 Mo group distributed their time across the whole maze, possibly because they were able to climb the maze wall, but the IL-12 Mo group had a higher density of activities in the lower-part area of the maze, which was in closer proximity to the start and the end points of the maze, whereas the PL-12 Mo group was more similar to the PL-1 Mo group in having their most activities in the opened segments of the maze, although the PL-12 Mo group was able to occasionally "break the barrier" to the unopened segment because they could climb the wall. These results demonstrate that the 12 Mo groups are different from their young counterparts in their learning progress and regularity, albeit the differences were mainly manifested in the IL groups.

During the memory retention test stage, the results of repeated-measurement ANOVA with age and learning mode as intergroup variables and session as an intragroup variable

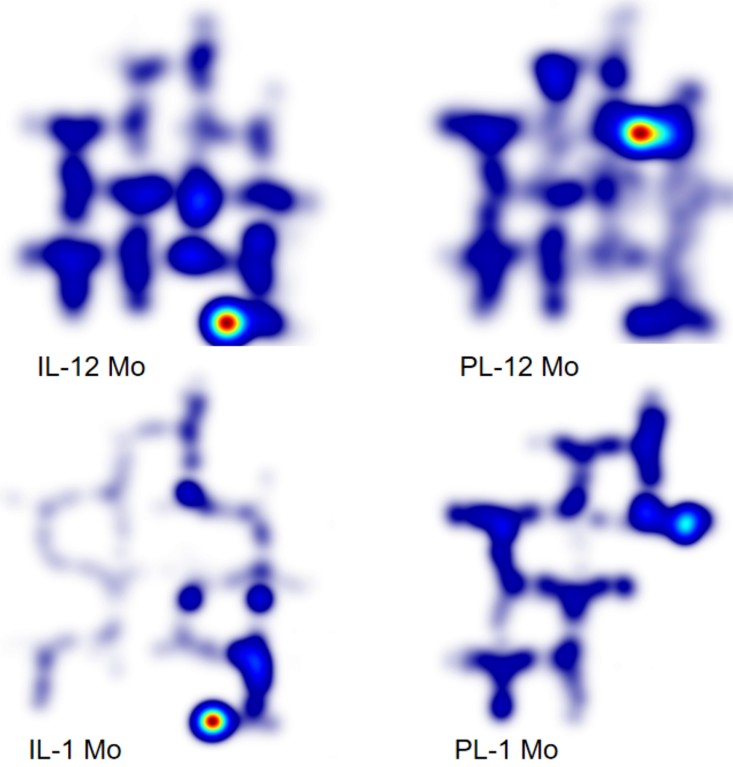

**Figure 5 Heatmap during Sub-stage Two of learning stage (Sessions 4–7).** During Sub-stage Two of learning stage, the IL-1 Mo group began to master the whole maze, as shown by the relatively light heat at the starting area and the relatively dense heat at the goal area; while the PL-1 Mo group only started to grasp the opened segments of the maze. The IL-12 Mo group, however, did not seem to have mastered the whole maze but explored mostly the shorter paths through climbing; and the PL-12 Mo group was also able to explore the whole maze through climbing but were mostly constrained in the opened segments.

show that the main effect of age was significant in the number of total errors made (Table 5). The young groups performed overall better than the 12 Mo groups, but the different learning modes had no effects on this stage (Fig. 4C).

In the Gestalt transfer learning task, the results of repeated-measurement ANOVA with age and learning mode as intergroup variables and session as an intragroup variable showed that the main effects of age and session were significant in the number of total errors made (Table 6). The 1 Mo groups performed overall better than the 12 Mo groups, though the differences between the two learning modes were not significant, mainly because unlike the 1 Mo groups, the IL-12 Mo group did not perform better than the PL-12 Mo group (Fig. 4D).

Results from an aggregated heatmap analysis for the 12 Mo groups throughout the experiments can be seen in Fig. 6. Few differences can be seen between the two learning modes during the memory retention test stage and the Gestalt transfer learning stage, albeit the obvious different distribution of activities between the two learning modes during the learning stage may reflect a slight advantage for the IL group.

**Table 5 Results of a three-factor repeated measures analysis of variance of the memory retention test stage after 1 week of learning.**

|  | SS | DF | MS | F (DFn, DFd) | p | $\eta^2$ |
|---|---|---|---|---|---|---|
| Session | 114.496 | 2 | 57.248 | $F(2, 62) = 1.782$ | 0.177 | 0.054 |
| Learning mode | 0.919 | 1 | 0.919 | $F(1, 31) = 0.033$ | 0.857 | 0.001 |
| Age | 1,183.928 | 1 | 1,183.928 | $F(1, 31) = 42.819$ | 0.000*** | 0.58 |
| Learning mode × Age | 6.14 | 1 | 6.14 | $F(1, 31) = 0.222$ | 0.641 | 0.007 |
| Session × Learning mode | 165.413 | 2 | 82.707 | $F(2, 62) = 2.575$ | 0.084 | 0.077 |
| Session × Age | 70.649 | 2 | 35.324 | $F(2, 62) = 1.100$ | 0.339 | 0.034 |
| Session × Learning mode × Age | 165.413 | 2 | 82.707 | $F(2, 62) = 2.575$ | 0.084 | 0.077 |

Note:
*** $p < 0.001$.

**Table 6 Results of a three-factor repeated measures analysis of variance of the number of total errors made in the Gestalt transfer learning stage.**

|  | SS | DF | MS | F (DFn, DFd) | p | $\eta^2$ |
|---|---|---|---|---|---|---|
| Session | 1,946.756 | 2 | 973.378 | $F(2, 62) = 16.823$ | 0.000*** | 0.352 |
| Learning mode | 4.53 | 1 | 4.53 | $F(1, 31) = 0.043$ | 0.836 | 0.001 |
| Age | 1,303.821 | 1 | 1,303.821 | $F(1, 31) = 12.487$ | 0.001*** | 0.287 |
| Learning mode × Age | 288.458 | 1 | 288.458 | $F(1, 31) = 2.763$ | 0.107 | 0.082 |
| Session × Learning mode | 162.289 | 2 | 81.145 | $F(2, 62) = 1.402$ | 0.254 | 0.043 |
| Session × Age | 34.848 | 2 | 17.424 | $F(2, 62) = 0.301$ | 0.741 | 0.01 |
| Session × Learning mode × Age | 185.902 | 2 | 92.951 | $F(2, 62) = 1.606$ | 0.209 | 0.049 |

Note:
*** $p < 0.001$.

In order to further elucidate the patterns, the number of total errors made during Sub-stage Three (Session 8–12) of the learning stage, memory retention test stage and Gestalt transfer learning stage was averaged for each rat (Fig. S1). Then non-repeated measure ANOVAs were conducted using the phases of the experiment as a within-subject factor, age and learning mode as between-subject factors (Table S5). The results show that the main effect of the phase of the experiment was significant ($F_{(2,62)} = 24.29$, $p < 0.001$, $\eta2 = 0.439$), and *post-hoc* multiple comparison tests reveal that the number of total errors made in the Gestalt transfer learning stage was significantly more than the other two phases of the experiment (Table S6). The main effect of age was also significant ($F_{(1,31)} = 45.139$, $p < 0.001$, $\eta2 = 0.593$), and *post-hoc* multiple comparison tests reveal that the number of total errors made by the 12 Mo groups was significantly more than that of the 1 Mo groups (Table S7). Further simple-effect analysis (Table S8) revealed that during Sub-stage Three (Session 8–12) of the learning stage, the PL-12 Mo group made significantly more total errors than the IL-12 Mo group ($F_{(1, 31)} = 11.01$, $p < 0.01$), whereas the IL-1 Mo group and the IL-12 Mo group did not have significant differences ($F_{(1, 31)} = 1.48$, $p = 0.233$). During the memory retention test stage, the two learning modes did not have significant differences between each other. During the Gestalt transfer
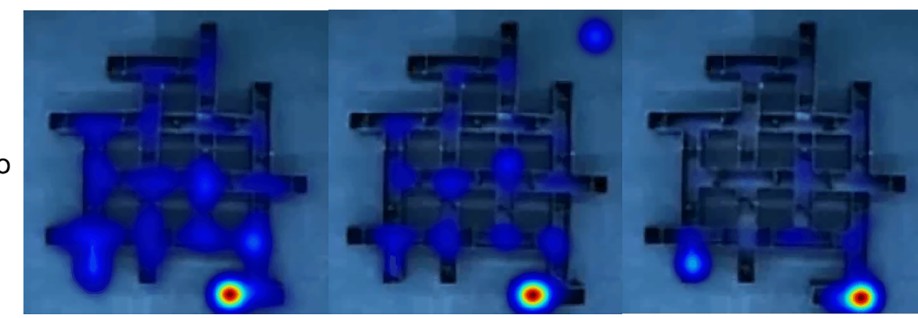

IL-12 Mo

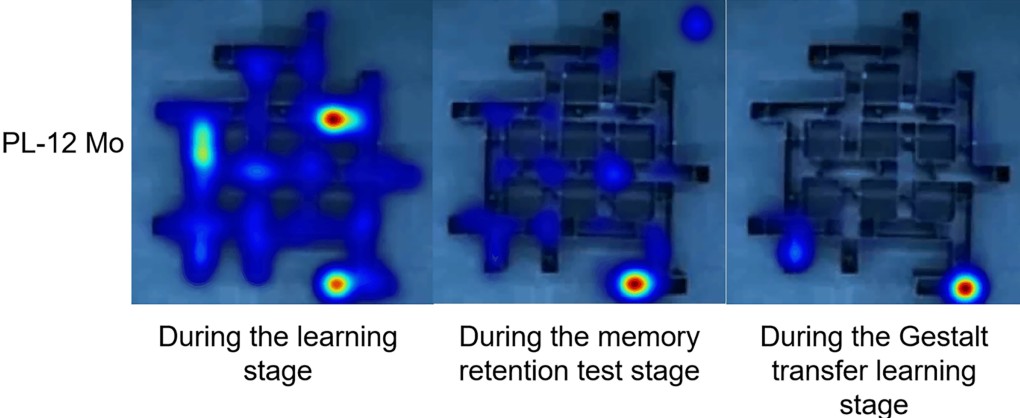

PL-12 Mo

| During the learning stage | During the memory retention test stage | During the Gestalt transfer learning stage |

**Figure 6 Aggregated heatmap across the tasks for the 12 Mo groups.**

learning stage, the two learning modes did not have significant differences between each other either.

## DISCUSSION

Maze learning in rats is a classic problem in the history of modern psychology (*Tolman & Honzik, 1930*; *Tsai, 1930*; *Buel, 1935*; *Honzik, 1936*; *Tolman, 1938*, *1948*; *Morris, 1984*; *Andrade et al., 2001*; *Dudchenko, 2001*). *Tsai (1930)* used to address the effects of two different instructional methods in maze learning experiments with rats, proving that repeated instructions could inhibit individual learning, which fundamentally challenged the relationship between teaching and learning. The result was cited by *Tolman (1938)* to overthrow the law of exercise—proposed by the well-respected educator and psychologist, Thorndike, who considered that any stimulus-response combination will gradually be strengthened during trial-and-error learning through regular practice and application. The impact of that experiment is still meaningful today (*Wang et al., 2021*). The difference between Tsai's experiment and ours is that Tsai found that adding more instructions was less effective while we found that adding more segmentation/limitation was less effective— we both found that whatever the experimenter as an instructor do to "help" the subject (the learner) to learn the task in an easier manner, it turns out to be a less effective one. Learning seems to ultimately depend on the integrative efforts made by the learner under a self-adapting mode, but not necessarily on the instructor[1]. From the viewpoint of the

[1] Therefore, the key role of the instructor may be better to switch to providing support for learners and stimulating their motivation for active learning.

neurobiology of learning, the results indicate the brain may not just be a learning machine that learns by simply strengthening corresponding synapses *via* repetition of the same information every day, nor does it learn by adding synaptic connections step by step—it may actually learn better by forming "schema" of the "whole" spectrum of information and then adjust the weights and connections of the synaptic network as it encounters stimuli (new and old) every day (*Tse et al., 2007*; *Farzanfar et al., 2022*).

While integrative learning was clearly advantageous in the 1 Mo groups (*Yin, Wu & Lian, 2020*), data from the current study suggest that such advantages may only present in the last segment of the learning stage in older rats but not in the Gestalt transfer learning stage. Was it due to differences in the learning process or was it because older rats had deteriorating memory systems to support higher-order functioning required by the IL mode? The heatmap analysis suggests that the differences do lie in the learning process especially for the IL-12 Mo group—indeed both old groups learned to climb the wall during the learning stage. *Kostić & Tošković (2022)* designed an experiment to determine which factor among path, time and effort was the key determinant of rodent behavior in exploring a maze to reach food, and they found that the subject always chose paths requiring less effort. Their findings provided a clue for why the older rats had attempted to climb the wall to reach the food-box—this may be less arduous for them compared with young rats. However, while climbing the wall might help reaching the food-box faster in some cases, it clearly slowed down the overall learning progress as compared to the 1 Mo groups. Nevertheless, the IL-12 Mo group may still benefit from the IL scheme from the beginning, resulting in a clear advantage over the PL-12 Mo group during the last segment of the learning stage, albeit this advantage did not extend further into the Gestalt transfer learning stage as in the 1 Mo groups, which suggest that the critical function supporting the linkage between learning and memory may be deteriorating in older rats.

It also worths noting that the IL groups did not simply have more repetitions of the whole route than the PL groups. First of all, the IL group and the PL group both had 12 sessions of learning, and for each session they could conduct free exploration within 15 min before reaching the foodbox and consuming food, upon which time the individual trial immediately ended and the subject was fetched out of the maze. Therefore, even though the PL group only had 5 days to experience the whole route, they had apprehended two thirds of the maze for 7 days and their total learning efforts (as reflected by the travel distance and the travel time) during Sub-Stage Three (Session 7–12) was much more than the IL group (Fig. S2) as the IL group quickly reached the foodbox with much fewer errors. Secondly, the IL group did not simply repeat during the 12 learning sessions. In fact, after they grasped the idea of the whole maze through initial exploration, they spontaneously segmented the whole route, focusing on learning the first "artificial"[2] segment to facilitate transferring of the memory (Fig. 5 of *Yin, Wu & Lian, 2020*), after which they started repetition to consolidate what they had learned through memory transfer to the less familiar parts of the routes (Fig. 5 of this article). Indeed, the segment fixation test of *Yin, Wu & Lian (2020)* further proved the objectivity of this memory transfer hypothesis, during which both the IL group and the PL group adopted the original route of the first segment and the new (direct) route of the second and the third segment, which suggest that

[2] We used the word "artificial" because there were indeed no segmenting barriers for the IL group.

the autonomous consolidation of the first segment for easier transfer later lie in the nature of learning. At the same time, the PL group was facing a new segment during the initial 1 or 2 sessions of each sub-stage, during which they focused on exploration but not exploitation —however, during the rest of each sub-stage the PL group also entered a repetitive mode, consolidating what they have learned, as reflected by the number of total errors made in Fig. 4A and session by session heatmap analysis (see Supplemental Data of this manuscript and *Yin, Wu & Lian, 2020*).

In sum, on one hand, the repeated learning of the PL group occurred 1 or 2 sessions right after a new segment was opened, while the repeated learning of the IL group occurred after consolidation of the "artificial" first segment in the three-segment maze—considering the actual travel distance and travel time they had in the maze for each session, the two groups only had substantial differences in when repeated learning occurred but did not have substantial differences in the total amount of repeated learning of the whole route. On the other hand, even if the whole route was opened to the IL group from the very beginning, the rats did not simply repeat the correct route and remember it, but instead they transferred the memory of a typical segment of the route to help learn the other segments to optimize the utilization of memory, which have been proved to be no different between IL and PL—the critical difference between the two learning modes, as we argue, lie in the fact that the IL group got to learn the whole route from the beginning and thus did the segmentation by themselves while the PL group's learning was constrained by pre-determined segmentation by the experimenter who might have thought that this would help the subjects learn better—this difference results in a different trajectory in how their memory is construed in their brain, as shown in Fig. 1.

This study is not without limitations—we could have designed a new maze with higher walls to avoid the problem of climbing in the 12 Mo groups—however, the high wall represents another technical challenge in tracking the subject that at the time we had no means to overcome. Therefore, as an informative study, we retain the original design of the apparatus to compare the effects of the different learning modes between the 1 Mo and the 12 Mo groups. Future studies could repeat our experiments in a more adapted apparatus that could accommodate subjects of different ages and sizes for confirmatory purposes. On the other hand, in this study we only had the 12 Mo male group to be compared with the 1 Mo male group due to the consideration that the high long-term maintenance cost of animals in our facility did not outweigh the potential benefits in answering research questions given that our previous study (*Yin, Wu & Lian, 2020*) found that the IL mode was more effective in the male group than in the female group. Future studies shall compare the effects of IL on the older female group with the older male group because older female rats may have different learning patterns from older males (*Stouffer & Barry, 2014*).

## CONCLUSIONS

In this study, we adapted the original design of *Yin, Wu & Lian (2020)* to investigate whether "Integrative Learning" was still advantageous in older rats. We found that the IL learning mode still promotes learning but not memory in older rats, possibly due to the

fact that those higher-order cognitive abilities that support meta-cognition, long-term retention and knowledge migration might be deteriorating in older rats. Further studies on neural correlates of such differences between the two learning modes in both young and older groups should provide us with further insights on the mechanisms and neural dynamics of such changes across different learning and memory stages. Furthermore, deep learning algorithms that feed on the real-time tracking points of the subjects under different learning modes could help formulate the theory of "Integrative Learning" in a way that it could provide more precise predictions for the relationship between learning mode and long-term outcome.

## ACKNOWLEDGEMENTS

We thank Ms. Si-Ping Cai and Ms. Jia-Wei Zhang for their assistance in maintaining and daily caring for the experimental subjects ever since they arrived at the animal facility. We also thank Ms. Ya-Xin Wang for her assistance in translating the names of the data files from Chinese into English.

### Funding

This work was supported by the key research project of the Ministry of Education of the People's Republic of China (16JJD190004); and the research start-up project of "Overseas Talents—Young Talents" of the Personnel Department of Fujian Normal University (Z0210509). The funders had no role in study design, data collection and analysis, decision to publish, or preparation of the manuscript.

### Grant Disclosures

The following grant information was disclosed by the authors:
Ministry of Education of the People's Republic of China: 16JJD190004.
Personnel Department of Fujian Normal University: Z0210509.

### Competing Interests

The authors declare that they have no competing interests.

### Author Contributions

- Bin Yin conceived and designed the experiments, performed the experiments, prepared figures and/or tables, authored or reviewed drafts of the article, and approved the final draft.
- Xiao-Rui Wu performed the experiments, analyzed the data, prepared figures and/or tables, authored or reviewed drafts of the article, and approved the final draft.
- Rong Lian conceived and designed the experiments, authored or reviewed drafts of the article, and approved the final draft.

## Animal Ethics

The following information was supplied relating to ethical approvals (*i.e.*, approving body and any reference numbers):

Institutional Animal Care and Use Committee of Fujian Normal University provided full approval for this research (Approval No. IACUC-20180019)

## Data Availability

The raw data are available in the Supplemental Files. The raw data contains all trials that were used for statistical analysis to compare the effects of IL and PL on 1 Mo and 12 Mo rats.

## Supplemental Information

Supplemental information for this article can be found online at http://dx.doi.org/10.7717/peerj.15101#supplemental-information.

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
