# Peer review of "“Integrative learning” promotes learning but not memory in older rats"

_PeerJ, doi:10.7717/peerj.15101_

## Round 0.1 · original submission · Major Revisions

Dear Drs. Yin and Lian,

Three experts have reviewed your work. Please make sure you address ALL concerns raised by these experts in your revision, especially when reporting details in experimental design and the validity of the results.

Thanks for your interest in submitting your work to PeerJ.

Hao Chen

·

Basic reporting

No comment

Experimental design

No comment

Validity of the findings

The description of the results is generally well written, adequate statistical analyzes were applied, the presentation of the results is satisfactory, the interpretation of the obtained findings is mostly adequate. However, I would suggest a few changes and additions:
1. In a sentence (lines 219 and 2020) “At the same time, the interaction between learning times and groups was significant.” It is confusing what exactly the variable "learning times" represents, I guess it should have been written "learning mode"; the same is true for a variable “groups”. My suggestion is to keep the same terminology for variables within each section of the manuscript. In addition, from graph 4a, we can see the average number of total errors for all groups during the first phase of the experiment, but the authors state in the text that there is an interaction between the two factors (learning times and groups), but they did not describe or show on the graph exactly what this interaction means. It is recommended to include the Post-Hoc test in the analysis of repeated measures and compare the obtained interaction at each level of the variable - sometimes it is necessary to add some syntax to SPSS for this data; for example:
/EMMEANS=TABLES(Learning mode * Age) COMPARE (Age) ADJ(BONFERRONI)
/EMMEANS=TABLES(Learning mode * Age) COMPARE (Learning mode) ADJ(BONFERRONI))

2. The next suggestion refers to the results related to Table 2. The way of listing the results is a bit unclear; namely finding written on this way “The number of days to learning success of the IL-young group was significantly shorter than that of the PL-young group, while the number of days to learning success of the old groups was generally more than that of the young groups.” (lines from 223 to 2025) it gives the impression that there is an interaction between learning mode and age, which was not obtained in the data. It seems that it would be clearer if the results were interpreted separately with regard to the main effects obtained; that is, which group needs fewer number of days to learning, older or younger rats; and also, which group needs fewer number of days to learning, Integrative Learning group or Progressive Learning group.

3. The following comment refers to the presentation of the obtained results from Table 3. Within the result display “However, unlike the young groups which showed significant differences between the IL mode and the PL mode all across three learning segments (Table 4), there was no significant differences between the IL mode and the PL mode in the old groups except for the last learning segment (Session 8-12), in which the IL mode showed an advantage by making significantly fewer total errors than the PL mode (Table 3).” (lines from 225 to 2030) the interaction obtained in sessions 1-3 (Session × Learning mode F (2, 26) = 6.504, p= .005) is not described.

The discussion is generally well written, but it would be good to amend and supplement if the recommendations for changes regarding the interpretation of the results are adopted.

Additional comments

It is not mandatory or necessary, more like a recommendation - I would recommend that the authors consider whether it is possible to average the number of errors for each rat for: three days during session 8 to 12, then the three-day during the retesting stage; and during Gestalt (reverse) transfer learning task; and then to do an analysis of variance where repeated measure will be the average number of errors in these three phases of the experiment, and non repeated measure age and learning mode.

Reviewer 2 ·

Basic reporting

In their manuscript “An animal behavioral model for the concept of “Integrative Learning”, Bin et al. examine the speed of learning a T-maze task by rats. In one group, they allowed the rats repeat the whole stretch of the maze from the first learning trial on 12 times (“integrative”). The other group was initially only presented the first third of the maze, then, after 3 trials, the first and the second stretch of the maze was presented; then, after another five trials, the whole maze was presented.
The methodology and the data are well presented in a professional English. The manuscript is well structured. The details of the experiments have been designed with great care; however, as explained below, there are major problems with the overall concept of the study.

Experimental design

The main result of the author’s study is that rats that have a chance to repeat a full maze course 12 times compared to rats that can repeat the full course of the maze only 5 times memorize the maze better. Moreover, rats in the second group have to relearn that their path is longer than in the initial trials. Thus, the interpretation of that simple finding as representing concepts of “integrative” vs “progressive” learning is to blow it out of proportion. Moreover, the way the authors drag parallels of their model to human learning is a gross overstatement. Overall, it seems that the concepts of “progressive” and “integrative” learning are neither models that will eventually result in improved efficacy of human learning, nor will it deepen our understanding of the biology of learning processes.
Overall, the manuscript must be completely rewritten. The authors must abstain from over-interpreting their data.

Validity of the findings

.

Additional comments

Typo Figure 1 legend, “all segments were kept open”
It is not clear why the reverse learning task is called “Gestalt” learning task, i.e. a reference for that terminology should be given.

·

Basic reporting

1. 12-month-old rats are considered middle-aged. PMID: 23930179, PMID: 20570402
2. It is really confusing for the reader to understand the study design, if not provide details. One solution is to add an illustration of the pipeline about the sessions.
3. "Sub-stage One, Two, Three" in the abstract were not consist throughout the manuscript.
4. Need language editing.

Experimental design

1. The authors tested their hypothesis on male rats, but not on female rats. Please explain the reason of excluding “older” females or add these data.
Older female rats may have different learning pattern vs. old males. PMID: 24122647
2. “travel distances and travel time” mentioned in the study design. However, I did not see any data analyzed in the result section.
3. When performing PL, where do you place the food box in session 4-7, and session 8-12?

Validity of the findings

1. Fig 5, I believe this heatmap is not about session 4-7. Session 4-7 is still the second segment training session for the PL. Therefore, there should be blank in the lower right corner in the PL-Old group.
2. According to results (see the tables 1-6), the learning mode did not make a big difference, but the age of rats did play a role in learning. How was the conclusion made for “Integrative Learning promotes learning but not memory in older rats”?

---

## Round 0.2 · Major Revisions

Dear Drs. Yin and Lian,

With the second round of review, the reviewers still have a wide range of recommendations. So I took it upon myself to read your article as well. On one hand, I do see the value of the new data that are included in the manuscript. On the other hand, I also agree with one of the reviewers that the manuscript overinterpreted the data, especially in the discussion section. You are more than welcome to submit a revised version, but please make sure the scope of introduction and discussion is limited to the parts most relevant to the experimental data that are presented.

Sincerely,

Hao Chen

·

Basic reporting

'no comment'

Experimental design

'no comment'

Validity of the findings

'no comment'

Additional comments

After revision the manuscript has been significantly improved so that no further changes are needed.

Reviewer 2 ·

Basic reporting

See initial review of the original manuscript and below "additional comments"

Experimental design

See initial review of the original manuscript and below "additional comments"

Validity of the findings

See initial review of the original manuscript and below "additional comments"

Additional comments

In the revised version of the manuscript the authors have not convincingly addressed the reviewer’s concern that a rather trivial finding (12 times repeating a complete task leads to better results than being allowed to repeat a task only 5 times) is much over-interpreted. The very fancy words and interpretation of the data sound initially interesting; upon deeper scrutiny they do not render much insight compared to what has been known for a long time.

·

Basic reporting

no comment

Experimental design

Regarding experimental procedures section (1), how to handle and interact rat for 15 mins per *cage*? and how to record the water intake and food intake for *individual* rat.

Validity of the findings

no comment

Additional comments

1. In Table S1 title, "Simple-simple-effect comparison" should be "Simple-effect comparison".
2. Please add "sprague-dawley (SD)" the first time it presents in the manuscript.

---

## Round 0.3 · accepted · Accept

Dear Dr. Yin,

Two reviewers now recommend the acceptance of this manuscript and I agree that most of the comments by the remaining reviewer have been mostly addressed. I, therefore, recommend the acceptance of this manuscript. Thank you for your effort in revising the manuscript and congratulations!

Hao

·

Basic reporting

no comment

Experimental design

no comment

Validity of the findings

no comment

Additional comments

no comment